# *Coxiella burnetii*: A Brief Summary of the Last Five Years of Its Presence in the Abruzzo and Molise Regions in Italy

**DOI:** 10.3390/ani14152248

**Published:** 2024-08-02

**Authors:** Alessandra Alessiani, Marco Di Domenico, Daniela Averaimo, Cinzia Pompilii, Marco Rulli, Antonio Cocco, Laura Lomellini, Antonio Coccaro, Maria Chiara Cantelmi, Carmine Merola, Elga Ersilia Tieri, Gianfranco Romeo, Barbara Secondini, Cristina Marfoglia, Giovanni Di Teodoro, Antonio Petrini

**Affiliations:** 1Istituto Zooprofilattico Sperimentale di Abruzzo e Molise “G. Caporale”, 64100 Teramo, Italy; m.didomenico@izs.it (M.D.D.); d.averaimo@izs.it (D.A.); c.pompilii@izs.it (C.P.); m.rulli@izs.it (M.R.); a.cocco@izs.it (A.C.); l.lomellini@izs.it (L.L.); a.coccaro@izs.it (A.C.); m.cantelmi@izs.it (M.C.C.); c.merola@izs.it (C.M.); e.tieri@izs.it (E.E.T.); g.romeo@izs.it (G.R.); b.secondini@izs.it (B.S.); c.marfoglia@izs.it (C.M.); g.diteodoro@izs.it (G.D.T.); a.petrini@izs.it (A.P.); 2Department of Bioscience and Technology for Food and Environment, University of Teramo, 64100 Teramo, Italy

**Keywords:** *Coxiella burnetii*, Q fever, sheep, goats, Italy

## Abstract

**Simple Summary:**

This study provides a description of the surveillance strategies adopted during an outbreak caused by the ST79 genotype of *Coxiella burnetii*, which has been endemic in the area for over a decade. It provides a detailed overview of *C. burnetii* infection in both wild and domestic animal populations in the Abruzzo and Molise regions. In the review of *C. burnetii* infections, data on the prevalence and impact of the bacterium in different animal hosts have been presented. This includes molecular findings that allow an assessment of the dynamics of infection before and after vaccination. In addition, this study proposes a novel NGS-based approach for in silico multi-spacer typing (MST) of *C. burnetii*. The results highlight the importance of surveillance and control measures in domestic animal populations to reduce the risk of zoonotic transmission.

**Abstract:**

*Coxiella burnetii* is the causative agent of Q fever. The main reservoirs for this bacterium, which can lead to human infection, in our region are typically cattle, goats, and sheep. In animals, *C. burnetii* infection is often detected due to reproductive problems. European Member States are required to report confirmed cases annually, but the lack of uniform reporting methods makes the data rather inconsistent. The Istituto Zooprofilattico Sperimentale dell’Abruzzo e del Molise is involved in official controls to identify the causes of abortions, monitor suspected or positive herds, evaluate suspected infections in pets and humans, monitor the spread in wildlife, etc. In this paper, we summarize the presence of *C. burnetii* over the last five years (2019–2023). Additionally, a detailed overview of *C. burnetii* infection in wild and domestic animals is provided. Five hundred sixty animals—including cattle; goats; sheep; wild animals, such as deer, boars, wolves, roe deer, owls, and otters; buffalo; dogs; horses; cats; and a donkey—and six human samples were tested by real-time PCR on the transposase gene IS1111 to detect *C. burnetii*. The MST profile was identified in some of the samples. Outbreaks of *C. burnetii* occurred in four herds. In one of them, it was possible to follow the outbreak from inception to eradication by evaluating the effect of vaccination on real-time PCR Ct values. A total of 116 animals tested positive for *C. burnetii*, including 73 goats, 42 sheep, and one bovine. None of the other samples tested positive. The strains for which the ST was performed were identified as ST79, a strain that has been present in the area for more than ten years. The effect of vaccination on the reduction of positive samples and the variation of real-time PCR Ct values was evaluated in strict correlation.

## 1. Introduction

*Coxiella burnetii* is an intracellular Gram-negative pathogen that usually causes late abortion, reproductive dysfunction, and subclinical infection in ruminants, while it is generally asymptomatic for all other animals. The large cell variant (LCV) of the bacterium corresponds to the intracellular replicative form, while the small cell variant (SCV) is the non-replicating form, released when the infected cells lyse, which can withstand a range of environmental stresses for long periods [1,2,3]. *C. burnetii* is a zoonotic agent able to cause Q fever in humans, which is a polymorphic disease ranging from acute symptoms (fever, hepatitis, pneumonia, etc.) to a chronic form with persistent focalized infections that lead to endocarditis, vascular infections, osteoarticular infections, and lymphadenitis [4,5,6,7,8]. The pathogen has a broad host range, but the main sources of human infection are cattle, goats, and sheep [9,10,11,12]. The pathogen largely spreads in the environment during parturition or abortion, and the main transmission route is aerogenic. It can also be shed in milk, vaginal mucus, urine, and feces. On the other hand, the role of ticks in human transmission and the oral route is still controversial. Q fever is potentially lethal [3,9,13,14,15,16,17], so much so that annual surveillance and confirmed cases reported by European Member States (MS) are required [18,19]. Acute Q fever mortality was assessed to be 1% in a study on hospitalized patients. The mortality in chronic cases has been reported as 13% and even up to 38% during an outbreak that occurred in the Netherlands between 2007 and 2010 [9,10]. The number of reports per 100,000 inhabitants in the EU/EEA was 0.17 in 2022. In 2022, Hungary had the highest notification rate, with 0.69 cases per 100,000 people, followed by Spain and Croatia, with 0.64 and 0.57 cases per 100,000 people [20]. Due to its characteristics, bacterial isolation is difficult, and the lack of standardized protocols makes comparison between results difficult. There are many discrepancies among the data obtained from serological or direct detection (molecular tests) to determine the *C. burnetii* prevalence. The data showed that in the last European One Health Zoonoses Summary Report (EUOHZ report), among animals using direct detection, infection rates were 3.6% for sheep, 2.1% for goats, and 2.7% for cattle. Among the herds tested, using direct detection methods, the proportion of positive herds was 4.9% for sheep, 3.5% for goats, and 7.7% for cattle. The proportion of seropositive herds was 97.7% for sheep and 30.1% for cattle, whereas no serological tests were reported for goat herds [20].

More stable and reproducible data are generated by the molecular typing method, multilocus spacer typing (MST). This tool was first described by Glazunova et al. (2005), who identified 10 highly variable intergenic spacers, and was then largely used for *C. burnetii* genotyping at high resolution [21]. The MST profiles described around the world are collected and shared through a public database (https://ifr48.timone.univ-mrs.fr/mst/coxiella_burnetii/ (accessed on 30 October 2023), allowing to infer their results in a global scenario. MST genotyping is indeed also considered a “geotyping” tool thanks to its ability to recognize specific epidemic clusters such as ST17 in French Guiana or ST33 in the Netherlands [4] but also other very limited local spots as previously described in the study areas ST78, ST79 and ST80 [3,22].

This study aimed to describe a full picture of *C. burnetii* detection in the Abruzzo and Molise regions in Italy based on real-time PCR results recorded by the laboratory of Diagnostica e Anatomo Istopatologia Unit of the Istituto Zooprofilattico Sperimentale dell’Abruzzo e del Molise (IZSAM) between 2019 and 2023. In this scenario, four different outbreaks that occurred in Abruzzo and the remerging of ST79 in a farm after ten years have been described [3,22].

## 2. Materials and Methods

### 2.1. Samples

Samples were collected as part of usual activity of the Istituto Zooprofilattico Sperimentale di Abruzzo e Molise (IZSAM) in the framework of official controls in order to identify the causes of abortions, monitor suspect or positive flocks, evaluate suspect infections of pets and humans, and monitor diffusion in wildlife animals [18,19,23]. No animals were killed for this purpose.

A total of 1041 samples, collected from 560 animal and 6 human specimens, were examined. The animals tested included 208 cattle (36.8%); 166 goats (29.3%); 97 sheep (17.1%); 41 wild animals, such as deer, boars, wolves, roe deer, owls, and otters (7.2%); 33 buffaloes (5.8%); 8 dogs (1.4%); 4 horses (0.7%); 2 cats (0.3%); and 1 donkey (0.2%). There were 6 humans tested (1.0%). The samples originated from carcasses or live animals (vaginal swabs and milk): a total of 294 came from spleens (28.2%), 248 from lungs (23.8%), 212 from milk (20.4%), 91 from livers (8.7%), 67 from blood (6.4%), 44 from vaginal swabs (4, 2%), 37 from brains (3.6%), 18 from placentae (1.7%), 16 from cotyledons (1.5%), 5 from fetal samples (0.5%), 4 from kidneys (0.4%), 2 from umbilical cords, 2 from lymph nodes (0.2% each), and 1 from abomasum (0.1%).

In particular, some of these samples belonged to four different farms (A, B, C, and D) located in Abruzzo.

Farm A was located in L’Aquila province; 71 goats were present on the farm. It delivered 2 fetuses in February 2023 (2 spleens, 2 lungs, and a placenta), with there being 63 samples (35 milk samples, 25 vaginal swabs, and 1 carcass including lung, liver, and spleen samples) at the end of the month and 44 samples from the same animals (milk) in May 2023. The herd was treated with Coxevac (Ceva Sante Animale, Libourne, France) in March, April, and September. In October, the breeder gave the last bulk milk sample. Two-sample *t*-tests were performed between Ct of real-time PCR-positive milk samples before and after vaccination to look for differences.

Farm B was located in Teramo province and had 43 sheep present on the farm. It gave 2 fetuses (2 spleens and 2 lungs) on the first of September 2023, 1 fetus (spleen, lung, and liver) after one week, and another fetus (lung and brain) the day after.

Farm C was located in Teramo province and had 23 sheep present on the farm. It gave 1 fetus in January 2022 (placentae and liver), 1 fetus (spleen and lung) and 1 vaginal swab in March 2022, and 17 vaginal swabs in June 2022.

Farm D was located in Pescara province and had 128 sheep and 2 cows present on the farm. It gave 2 fetuses in September 2023 (2 spleen, 2 lungs, 1 cotyledon), 3 milk samples in November, and 32 milk samples in December 2023, all from sheep.

### 2.2. DNA Extraction and Real-Time PCR

DNA from tissues and vaginal swabs was extracted using Maxwell 16 Tissue DNA Purification Kit (Promega, Milano, Italy) according to the manufacturer’s instructions. Cells from milk were obtained by centrifugation of 50 mL at 2000× *g* for 10 min; the pellet was resuspended with 300 µL of nuclease-free water and extracted using Maxwell 16 Cell DNA Purification Kit (Promega, Italy) according to the manufacturer’s instructions. All samples were tested by real-time PCR to detect the presence of transposase gene IS1111 of *C. burnetii,* as previously described [3,22]. Briefly, the primers and probe were as follows: Forward CoxbS: 5′-GATAGCCCGATAAGCATCAAC-3 [300 nM]; Reverse CoxbAs: 5′-GCATTCGTATATCCGGCATC-3′ [300 nM]; and Cox probe: 5′-6-FAM-TGCATAATTCATCAAGGCACCAATGGT-TAMRA-3′ [100 nM] (Eurofins, Cuneo, Italy). The 20 µL reaction mixture contained the following: 10 µL of TaqMan Fast Universal PCR Master Mix (Thermo Fisher Scientific, Carlsbad, CA, USA), 0.2 µL of probe, 0.2 µL of each primer, 3.6 µL of nuclease-free water (Promega, Italy), and 5 µL of DNA. Real-time PCR was performed by Quant Studio 7 Pro Real-Time PCR system (Applied Biosystems, Monza, Italy). Amplification protocol performed was 20 min at 95 °C followed by 40 cycles of 1 s at 95 °C and 20 s at 60 °C.

### 2.3. Molecular Characterization

Molecular characterization of *C. burnetii* DNA of Farm A samples was performed as described by Glazunova et al. (2005) with some modifications [3,21]. Each PCR was carried out in a Gene Amp PCR System 9700 (Applied Biosystems, Monza, Italy). Five microliters of the extracted DNA were amplified in a 50 µL reaction mixture containing 200 nM of each primer, 200 mM dNTPs (Promega, Madison, WI, USA), 2.5 mM MgCl_2_ Solution, 0.03 U/µL Ampli Taq Gold, and 1X PCR Buffer II (Applied Biosystems, Monza, Italy). Amplifications were carried out under the following conditions: initial denaturation of 10 min at 95 °C, followed by 40 cycles of denaturation for 30 s at 95 °C, annealing for 30 s at 57 °C, and extension for 30 s at 72 °C. Final extension was performed at 72 °C for 7 min. PCR products were purified using the QIAquick PCR Purification Kit (Qiagen, Milano, Italy) and sequenced by NextSeq2000 (Illumina, San Diego, CA, USA). Briefly, PCR-purified products were pooled equivolume and processed using the Illumina DNA Prep protocol. Fastq files were then analyzed through the NGS manager “https://github.com/genpat-it/ngsmanager (accesed on 30 October 2023)”, the bioinformatics pipeline of the GENPAT platform “https://github.com/genpat-it (accessed on 30 October 2023)”, and the Italian national platform for collection, analysis, and sharing microbial pathogen’s WGS data. In detail, raw reads were trimmed using fastp (V0.23.1) before host depletion (goat) through bowtie2 (v2.5.3). Finally, de novo assembly was obtained using shovill (v1.1.0). Contigs were then uploaded to the MST database for sequence comparison “https://ifr48.timone.univ-mrs.fr/mst/coxiella_burnetii/blast.html (accessed on 30 October 2023)” and allele assignment.

## 3. Results

### 3.1. Samples and Real-Time PCR

In general, *C. burnetii* DNA was detected in 116 animals (20.7%): 73 goats (13%), 42 sheep (7.5%), and one cow (0.17%). Four hundred forty-four (79.3%) animals (domestic and wild) were negative. There were 126 (12.1%) positive real-time PCR samples, consisting of 63 milk samples (50.0%), 29 vaginal swabs (23.0%), 10 lungs and 10 livers (0.8% each), six spleens (4.8%), four placentae (3.2%), and four other samples (3.2%). In 2023, the highest positivity rate was detected: 62 out of 99 goats (62.6%) and 21 out of 24 sheep (87.5%). A total of 915 (87.9%) samples were negative. The distributions of positive samples per year and the regions they were found in were as follows: one in 2019 in Molise, zero in 2020, four in 2021 in Abruzzo, fifteen in 2022 in Abruzzo, one hundred five in Abruzzo, and one in Molise through 2023. 

Regarding the samples on the farms of interest, the following is noteworthy:

Farm A: A total of 113 samples, consisting of 77 positive and 36 negative samples, were examined. One placenta tested positive the first time; 35 milk samples, 25 vaginal swabs (some from the same animal), and one lung tested positive the second time. After the second vaccination, 15 milk samples tested positive. In October 2023, the breeder transferred the last milk sample, which was negative. The prevalence observed at peak times was 75%.

For this farm, we were able to evaluate the results of the relation relationship between the sampled milk before and after vaccination. There were 35 and 15 positive milk samples for the first and second samplings, respectively. 

The mean Ct values of milk were 31.75 before vaccination and 39.13 after vaccination. *t*-tests between the real-time PCR Ct of sampled milk before and after vaccination of the entire herd showed a significant difference between the two sets of values (*p* < 0.05).

Farm B. All nine samples (four lungs, three spleens, one brain, and one liver) tested were positive. The prevalence observed at peak times was 5%.

Farm C. There were 21 samples tested, consisting of six positive (four vaginal swabs, one liver, and one placenta) and fifteen negative samples (13 vaginal swabs, one lung, and one spleen). The prevalence observed at peak times was 17%.

Farm D. There were 40 samples tested, consisting of 12 positive (11 milk samples and one lung) and 28 negative samples (24 milk samples, two spleens, one lung, and one cotyledon). The prevalence observed at peak times was 7%.

Table 1 shows the number of positive animals on tested animals at each moment of delivery in IZSAM from the farms of interest (A, B, C, D) and the total animal number in the farm.

### 3.2. Molecular Characterization

The NGS manager workflow released a multifasta file containing ten contigs corresponding to the MST loci. The allelic combination obtained blasting the ten marker sequences to the reference database was 5 4 2 5 1 5 3 2 4 4, which is currently coded as MST 79 in the group list “https://ifr48.timone.univ-mrs.fr/mst/coxiella_burnetii/groups.html (accessed on 30 October 2023)”. This MST profile is similar to the widespread ST8 but different at Cox56, showing allele 2 instead of 3. These two alleles differ in the deletion of three nucleotides (ATG) at position 318–320.

## 4. Discussion

The *C. burnetii* detection rate in animals tested in the Abruzzo and Molise regions ranged from 4.2% to 87.5% in sheep (87.5% in 2023) and from 7.1% to 62.6% in goats (62.6% in 2023), while in cattle, the detection rate was 1.2% in 2023. Sheep, goats, and cattle were the main tested species, as was the case in other European countries [5,18]. Nevertheless, a significant number of wild animals and buffaloes contributed to consistent results regarding *C. burnetii* distribution in the study area. Samples were collected from animals potentially infected with *C. burnetii* due to abortion (sheep, goats, cattle, buffaloes, etc.) or herds that were monitored with a history of positive cases. They originated mainly from carcasses, abortion products, and milk.

The infection rate, except for cattle, was higher in 2023 than in the EUOHZ zoonoses report: from 3.6 to 18.3% in sheep (3.6% in 2022), from 2.1 to 16.5% in goats (2.1% in 2022), and from 2.7 to 5.1% in cattle (2.7% in 2022). These data are, however, quite different from those indicated by a study conducted in Sardinia, where the prevalence in 2018 was around 34% in small ruminants [24]. Also, the percentage increase observed in sheep and goats is in contrast to the inverted trend over the years described in other countries in zoonoses reports. This reflects the high rate of focalized sampling from suspected or confirmed *C. burnetii*-infected flocks. Conversely, data observed in cattle are in line with the EUOHZ report [18].

Four different suspected outbreaks in the Abruzzo region have been described in this study in the past two years. The animals involved were goats on Farm A and sheep on other farms. None cohabitate with other livestock species except Farm D, where there are two cows. There is no evidence of domestic animals, such as dogs and cats, roaming around the stables. No acute human cases have been reported during the referred period. No serological analysis has been carried out on breeders or farm veterinarians. The samples have been collected for institutional surveillance, such as passive surveillance of “category E” pathogens [18,19,20]. In just one case has it been possible to follow the course of the outbreak from the onset to eradication to investigate characteristics related to treatment and genetic compounds. All index cases showed abortion, like the first symptoms, which led the breeder to confer the sample on IZSAM. Within farms, the highest prevalence observed at peak times was 75% in Farm A, while the lowest was 5% in Farm B. Probably all of them were able to create an outbreak among the humans. The last outbreak detected in Italy in 2021, which involved both humans and animals (cows), showed a prevalence of 14% in the animal population, and again, the cattle that caused the infection did so during calving, spreading the bacterium through the air [25]. Real-time PCR detection of IS1111 in milk and vaginal swabs was a very useful indicator of herd infection status. In particular, it was possible to estimate the response of vaccinated animals over time. Despite some animals showing no change in *Coxiella* elimination, the difference in mean Ct values of real-time PCR before and after vaccination is evaluable, indicating that the treatment worked even during infection. These results confirmed previous data in both goats and cows treated with phase I vaccines [26,27,28,29,30]. The possible cause of this variability could depend on the individual’s response to vaccination, provided that the vaccination procedure was carried out well and on all animals on the farm (this could also be an explanation).

At least five different genotypes have been described in the study area [3,22], which are currently reported in the updated database as ST32, ST55, ST78, ST79, and ST80 “https://ifr48.timone.univ-mrs.fr/mst/coxiella_burnetii/groups.html (accessed on 30 October 2023)”. ST32 harbors the QpH1 plasmid associated with endocarditis in France [9] and is frequently found in both goats and sheep in Southern Europe [31]. ST55 is a less frequent genotype harboring the QpRS plasmid significantly associated with persistent focalized infections [9]. It is closely related to ST8 and its variant ST79, which are only described in this area, as well as ST78 and ST80. The genotype ST78 clusters together with ST32, while the closest genotype of ST80 is ST27, for which there are no data about the plasmid type. Mapping circulating genotypes in a selected area is helpful in predicting and contrasting pathogens spread even through animal products (e.g., milk and cheese). Indeed, the ST12 and ST32 found in cheese samples in Tuscany [32] indicate the presence of these genotypes in herds of the same region [22] in a large period considered.

The ST79 genotype, identified in goat samples from farm A, appears to be endemic to our region. It was first described in 2013 and two years later, in 2015, in both goats and sheep and was formerly named the ST8 variant [3,22]. This is an interesting situation because, although there are endemic strains worldwide, the enormous plasticity of the *C. burnetii* genome made this finding, over ten years, peculiar. Perhaps the sampling of soil and dust from the interested farm or from nearby pastures, as appropriate samples to detect *C. burnetii* in the environment [33] would have made it possible to trace the presence of ST79 in the specified area over time. The significance of infection from material deposited in the environment versus transmission directly from infected animals is not known. In the study of Kersh (Kersh et al., 2013), it was supposed that the highest concentrations of environmental *C. burnetii* are found in goat birthing areas and that contamination of other areas is mostly associated with human movement [6]. Eldin (Eldin et al., 2013) connected the environmental resistance with SCVs’ cell form. Because they are highly resistant to many stresses, they can survive for 7 to 10 months on wool at ambient temperature, for more than 1 month on fresh meat, and for more than 40 months in milk [34].

In 2022, the number of Q fever human cases increased compared to the 2020–2021 pandemic years but was similar to that reported in the pre-pandemic period. France, Germany, Hungary, and Spain accounted for about 78.2% of cases. An outbreak of airborne Q fever (20 confirmed cases) was reported in Croatia and was associated with non-occupational exposure in a semi-urban area of Čavle due to environmental contamination at three farms. However, the trend of Q fever for the period 2018–2022 did not show a significant increase or decrease in the EU. In 2022, the percentage of fatal cases showed a decrease (0.90%) compared to 2021 (1.5%) [20].

In Italy, in 2003, 133 cases of acute Q fever were recorded in the Prevention Department, Azienda Sanitaria Locale (ASL) of Como, presenting with high fever, dry cough, arthralgia and fatigue, and atypical pneumonia. Of these, 59 individuals were inmates from local prisons (which are located close to pastures grazed by sheep that were examined, and the prevalence of the disease was 34.2%), 37 prison officers, 33 people living in the area traveled by the herd, and four were staff of veterinary services [35,36]. The last detected outbreak was in 2021. The outbreak originated from parturition by an infected cow, with the spread of *C. burnetii* by air and infection via the respiratory route. Four people were hospitalized, but there were no deaths [25]. The absence of acute clinical manifestations in the Abruzzo and Molise areas may be related to the presence of the ST79 strain, which is closely related to ST8 and other strains (e.g., 6, 9, 55) that harbor the QpRs plasmid. This mobile genetic element was only described in patients with persistent focalized infections and was never associated with any acute Q Fever strain [13]. Anyway, this argumentation could be mere speculation, and human cases cannot be excluded; indeed, despite the increasing attention to Q-fever in the last decade, the disease is still widely underestimated and underdiagnosed [37,38].

## 5. Conclusions

Data collection following five years of passive surveillance of *Coxiella burnetii* in the Abruzzo and Molise regions led to several important considerations about the type and efficacy of surveillance employed. From 2019 to 2023, an increasing prevalence of *C*. *burnetii* was observed in this study. This rise was primarily the result of the diligent application of surveillance plans and the accurate abortion cause diagnosis, benefiting both breeders and consumers. However, to gain a more comprehensive understanding and control over this zoonotic pathogen, it is necessary to apply the same rigorous surveillance criteria on a larger scale and in a uniform manner.

The current surveillance results from the EUOHZ initiative are insufficient to analyze trends for Q fever at the EU level. Despite the commencement of annual EU reporting in 2021, which classified Q fever as a ‘Category E’ disease, this initiative had little impact on the consistency and comparability of reporting data between countries [20]. The discrepancies in the results reported by different Member States (MS) and non-Member States are primarily due to variations in sampling strategies, such as the types of samples collected, the testing methods used, the coverage of surveillance, data completeness, and the sensitivity of the surveillance methods.

Direct detection of the pathogen’s DNA through molecular methods seems to be a more effective approach for surveillance. For a more accurate depiction of *C. burnetii* circulation within animal populations, which could serve as a dangerous source of infection for humans and result in economic damage for farmers, active surveillance should be considered. This active surveillance could focus on matrices such as milk or vaginal swabs. This type of surveillance could offer a clearer picture of *C. burnetii* distribution. For example, analyzing milk and vaginal swabs could yield critical insights into infection dynamics within herds. This approach could also aid in early detection of outbreaks, thereby preventing transmission to humans.

Vaccination has also proven to be an efficient method to contain the spread of *C. burnetii*, as demonstrated by previous experiences in the Netherlands. Implementing vaccination programs could significantly reduce the prevalence of this pathogen in animal populations. Moreover, harmonization of detection methods at both the national and European levels is crucial. Such harmonization would be key to clarifying the real prevalence of *C. burnetii* across different territories, ensuring that data are comparable and reliable.

The observed increase in prevalence in the Abruzzo and Molise regions during the five-year study period highlights the urgent need to strengthen and standardize surveillance practices. This would not only protect public health but also safeguard the economic interests of breeders. Passive surveillance, while useful, has limitations in sensitivity and specificity in detecting infections. Therefore, adopting molecular methods for the detection of pathogen DNA could significantly enhance the ability to identify the presence of *C. burnetii* in animals.

In conclusion, the data collected from passive surveillance of *C. burnetii* in Abruzzo and Molise emphasize the need for a more integrated and uniform approach to managing Q fever. Adopting molecular methods, enhancing active surveillance, implementing vaccination programs, and harmonizing surveillance practices at the European level are critical steps. These measures will not only protect public health but also promote economic welfare for breeders, contributing to more effective and sustainable management of both animal and human health resources.

## Figures and Tables

**Table 1 animals-14-02248-t001:** Period of analysis, composition of the herd, and positive animals among tested animals for each farm.

Farm	Period	Animal in Farm	Positive Animals for Time 0/Tested Animals	Positive Animals for Time 1/Tested Animals	Positive Animals for Time 2/Tested Animals	Vaccination
**A**	FEB 2023/MAY 2023	71 GOATS	1/1	53/53	15/44	YES
**B**	SEPT 2023	43 SHEEP	2/2	1/1	1/1	NO
**C**	FEB 2022/JUN 2022	23 SHEEP	1/1	4/4	N.C.	NO
**D**	SEPT 2023/DEC 2023	128 SHEEP	1/2	1/1	9/32	NO

## Data Availability

The original contributions presented in the study are included in the article.

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
