# Peer review of "Coxiella burnetii: A Brief Summary of the Last Five Years of Its Presence in the Abruzzo and Molise Regions in Italy"

_animals, 2024, doi:10.3390/ani14152248_

Round 1

Reviewer 1 Report

Comments and Suggestions for Authors

Review for Animals (June 2024)

The article entitled “Coxiella burnetii: a brief summary of the last five years of presence in central Italy regions” provides data about infection with this pathogen in animals in two Italian regions. The Authors present results of real-time PCR analyses, focusing on samples from domestic ruminants from 4 farms. Moreover, they successfully genotyped the samples from one of the farms, providing crucial data about genotyping characterisation of the strains present in the region.

General comments:

The manuscript is consistent, well-written and easy to follow in the majority of sections. However, some improvements should be implement to Materials and Methods and Results sections. It is difficult to follow and connect information about the samples with data present in results section, because numbers of tested samples present in Results are incongruent with numbers mentioned in Materials and methods. It might be caused by the fact that fetuses or carcass are in one point present as a samples, but it can be assumed from the context that their internal organs were treated as separate samples. This issue needs to be clarified and harmonised by the Authors. Moreover, numerous species are mentioned as tested in “Samples” section, although no data about results are present.

The methods are described clearly and adequately with one exception. The results of statistical analysis are presented, although there is no information about statistical analyses in Materials and methods section. The Authors should add an appropriate description in aforementioned section. The statistic values were calculated based on Ct values that are not present in the manuscript. The Authors should provide the availability of this data for the Readers.

Detailed comments

Lines 11-12: The reviewer would be grateful if the Authors explain where “a comprehensive description of the management strategies adopted during an outbreak caused by ST79 genotype” are presented in the manuscript.

Lines 13-14: The Authors stated that the manuscript “(…)provides a detailed overview of C. burnetii infection in both wild and domestic animal population”, although the data about wild animals are limited only to number of tested samples without any details about tested species and type of tested specimens collected from them as well as the results.

Line 30 – Please see comment for lines 13-14.

Line 53-54 - The Authors should consider recalling the data about number of reports per 100,000 inhabitants in EU/EEA at least for 2022, especially that they present the data for 2022 for some countries.

Line 59: The abbreviation EUOHZ should be explained as it is used in the text for the first time.

Materials and Methods

Lines 90-91: The Authors tested 560 animals and a majority of tested specimens were internal organs (294 spleens, 248 lungs, 91 livers). It would be more informative for readers if the Authors clarify why these types of samples were dominated and what were the sources (aborted fetuses?). Moreover if the Authors would like to provide "a detailed overview of C. burnetii infection in both wild and domestic animal population" presenting of the data about types of specimens tested from individual animal species should be considered.

Lines 96-106 - Fetuses are mentioned as tested specimens, but it is incongruent with the list of samples presented in lines 90-93 (fetuses are not listed as samples). Assuming that some internal organs were tested as separate samples, it should be clarified and harmonised by the Authors.

Line 97 – The carcass is mentioned as one of tested specimens, but it is incongruent with list of specimens presented in lines 90-93 (carcass is not mentioned). Assuming that some internal organs of carcass were tested as separate samples, it should be clarified by the Authors.

Line 97 -  It would be more informative for readers if the Authors specify how many of the 53 tested specimens were milk and swabs.

Line 105-106 – The Authors should clarify from what animal species were obtained the samples as cattle and sheep were raised in the farm.

Results

Line 145: The percentage of positive animals should be calculated excluding human samples (116 divided by 560 (not 566) -  is 20.71).

Data about numbers of tested samples collected from many species are given in “Samples” subsection, although the results of testing are present only for goats, cattle and sheep. The Authors should clarify the results obtained for other species (e.g. by mentioning that all specimens collected from them were negative (?) in real-time PCR). 

A 104 of 126 positive samples were collected from the 4 Farms (A to D).  It is difficult to deduce from the text about the rest of positive samples. What type of specimens were they and from what species were they collected? The Authors should consider specifying the details about the rest of positive specimens (i.e. location of the farm etc.) as it would be interesting for readers. A map or table summarizing the details would be useful.

Line 146-150 – The percentages of positive milk/swabs/lungs etc. samples among all tested samples were calculated. The Authors should consider extending the results through estimating the percentage of positives for certain type of samples (i.e positive milk specimens/all tested milk specimens).

Line 151 – Number of tested samples (n=113) is incongruent with number presented in point 2.1. Samples. In lines 96-100 it is stated that 2 fetuses, 53 samples (milk swab and carcass), 44 milk samples and 1 BTM were tested (in total 100 samples). Assuming that some internal organs of foetuses were tested as separate samples, it should be clarified and harmonised by the Authors.

Line 152 – The reviewer would be grateful to the Authors for explaining the term “conferred milk” in this context

Line 155 – The results of T test are presented, although there is no information about statistical analyses in Materials and methods section. The Authors should add an appropriate description in aforementioned section. The T values were calculated based on Ct values that are not present in the manuscript. The Authors should provide the availability of this data for the Readers.

Line 161 - The authors should explain how many samples were tested. It is stated that 35 milk and 25 swabs were tested the second time, whereas, according to data from line 97, there were only 53 samples (milk, swabs and carcass) collected the second time (?).

Line 165: As it was addressed in the previous comments: In point 2.1 “Sampling” the Authors present number of fetuses tested from the farm (4 animals), but no data about number or type of tested samples collected from this fetuses are given.  The result “All 9 samples tested were positive” is non-informative, because it is unknown what type of samples were collected and tested and if they were collected from each fetus.

Line 167 – The Authors should clarify what 21 samples were tested - the data in lines 103-104 indicates that 1 fetus and 4 swabs were obtained for testing from this farm. It can be assumed that few samples were tested from fetus, but a difference of 15 samples seems too big.

Line 169 - As it was addressed in the previous comments: Number of tested samples (n=46) is incongruent with number presented in subsection “Samples”. In lines 105-106 it is stated that 2 fetuses and 33 milk samples were tested. Assuming that some internal organs of fetuses were tested as separate samples, it should be clarified and harmonised by the Authors.

Line 175 Table 1 – There is no reference to the table in the text. Were animals from farm A treated using drugs? If they were vaccinated, as it was mentioned in the text, the Authors should rephrase the heading of column to “Vaccination”. Moreover, it is not clear how many cows and sheep from Farm D were tested and were positive.

Line 201: The Authors should check if the number of reference is correct. It seems like it should be 20, not 18.

Line 207: The grammar should be corrected (e.g. has been done)

Line 256-258: The sentence is unclear and should be rephrased.

Line 284: It is concluded that “ From 2019 to 2023 an increasing prevalence of C. burnetii was observed in this study”. Unfortunately, no results confirming this trend were present by the Authors in Result section. Only the calculations for 2023 were specified.

Lines 327-330 – The Authors should consider combining this paragraph with sentence in lines 304-305 as it seems to be a continuation of the same thought.

Lines 331-332 – The Authors should consider moving this sentence to paragraph where the vaccination was discussed (lines 306-312).

References should be checked adjusted to the requirements of the Journal (e.g reference number 5 is given as: “ GACHE, K. and ROUSSET, E. and PERRIN, J. B. and DE CREMOUX, R. and HOSTEING, S. and JOURDAIN, E. and GUATTEO, 363

R. and NICOLLET, P. and TOURATIER, A. and CALAVAS, D. and et al. Estimation of the frequency of Q fever in sheep, goat 364

and cattle herds in France: results of a 3-year study of the seroprevalence of Q fever and excretion level of Coxiella burnetii in 365

abortive episodes 366

. Epidemiology and Infection:3131–3142”)

Author Response

Comment 1: The manuscript is consistent, well-written and easy to follow in the majority of sections. However, some improvements should be implement to Materials and Methods and Results sections. It is difficult to follow and connect information about the samples with data present in results section, because numbers of tested samples present in Results are incongruent with numbers mentioned in Materials and methods. It might be caused by the fact that fetuses or carcass are in one point present as a samples, but it can be assumed from the context that their internal organs were treated as separate samples. This issue needs to be clarified and harmonised by the Authors. Moreover, numerous species are mentioned as tested in “Samples” section, although no data about results are present.

The methods are described clearly and adequately with one exception. The results of statistical analysis are presented, although there is no information about statistical analyses in Materials and methods section. The Authors should add an appropriate description in aforementioned section. The statistic values were calculated based on Ct values that are not present in the manuscript. The Authors should provide the availability of this data for the Readers.

Response: thank you very much for your valuable and constructive reviews. In general, we have extensively revised the indicated sections, making the article more congruent and fluid. The Ct values are inserted.

Comment 2: Lines 11-12: The reviewer would be grateful if the Authors explain where “a comprehensive description of the management strategies adopted during an outbreak caused by ST79 genotype” are presented in the manuscript.

Response 2: The sentence has been appropriately changed to describe the work more appropriately.

Comment 3: Lines 13-14: The Authors stated that the manuscript “(…)provides a detailed overview of C. burnetii infection in both wild and domestic animal population”, although the data about wild animals are limited only to number of tested samples without any details about tested species and type of tested specimens collected from them as well as the results. 

Response 3: The wild animals were all negative, so as not to give too much information, they were not described specifically. However, a small description has been added.

Comment 4:Line 53-54 - The Authors should consider recalling the data about number of reports per 100,000 inhabitants in EU/EEA at least for 2022, especially that they present the data for 2022 for some countries. 

Response 4: Done

Comment 5: Line 59: The abbreviation EUOHZ should be explained as it is used in the text for the first time.

Response 5: Done

Comment 6: Lines 90-91: The Authors tested 560 animals and a majority of tested specimens were internal organs (294 spleens, 248 lungs, 91 livers). It would be more informative for readers if the Authors clarify why these types of samples were dominated and what were the sources (aborted fetuses?). Moreover if the Authors would like to provide "a detailed overview of C. burnetii infection in both wild and domestic animal population" presenting of the data about types of specimens tested from individual animal species should be considered.

Response 6: The text was modified to  better explane the type of samples.

Comment 7: Lines 96-106 - Fetuses are mentioned as tested specimens, but it is incongruent with the list of samples presented in lines 90-93 (fetuses are not listed as samples). Assuming that some internal organs were tested as separate samples, it should be clarified and harmonised by the Authors.

Line 97 – The carcass is mentioned as one of tested specimens, but it is incongruent with list of specimens presented in lines 90-93 (carcass is not mentioned). Assuming that some internal organs of carcass were tested as separate samples, it should be clarified by the Authors.

Line 97 - It would be more informative for readers if the Authors specify how many of the 53 tested specimens were milk and swabs.

Line 105-106 – The Authors should clarify from what animal species were obtained the samples as cattle and sheep were raised in the farm.

Response 7: The entire section has been rewritten and the data about the samples have been corrected.

Comment 8:  Line 145: The percentage of positive animals should be calculated excluding human samples (116 divided by 560 (not 566) - is 20.71).

Data about numbers of tested samples collected from many species are given in “Samples” subsection, although the results of testing are present only for goats, cattle and sheep. The Authors should clarify the results obtained for other species (e.g. by mentioning that all specimens collected from them were negative (?) in real-time PCR). 

A 104 of 126 positive samples were collected from the 4 Farms (A to D). It is difficult to deduce from the text about the rest of positive samples. What type of specimens were they and from what species were they collected? The Authors should consider specifying the details about the rest of positive specimens (i.e. location of the farm etc.) as it would be interesting for readers. A map or table summarizing the details would be useful.

Response 8: All the data have been corrected and the text have been rewritten to better explain the results.

Comment 9:Line 146-150 – The percentages of positive milk/swabs/lungs etc. samples among all tested samples were calculated. The Authors should consider extending the results through estimating the percentage of positives for certain type of samples (i.e positive milk specimens/all tested milk specimens). 

Response 9: the data would certainly be interesting, but given the large number of percentages expressed, it could create reading difficulties without adding much to the description of the situation.

Comment 10: Line 151 – Number of tested samples (n=113) is incongruent with number presented in point 2.1. Samples. In lines 96-100 it is stated that 2 fetuses, 53 samples (milk swab and carcass), 44 milk samples and 1 BTM were tested (in total 100 samples). Assuming that some internal organs of foetuses were tested as separate samples, it should be clarified and harmonised by the Authors.

Response 10: The section has been rewritten to better explain itself.

Comment 11: Line 152 – The reviewer would be grateful to the Authors for explaining the term “conferred milk” in this context

Response 11: The sentence has been rewritten

Comment 12: Line 155 – The results of T test are presented, although there is no information about statistical analyses in Materials and methods section. The Authors should add an appropriate description in aforementioned section. The T values were calculated based on Ct values that are not present in the manuscript. The Authors should provide the availability of this data for the Readers.

Response 12: the type of t test has been inserted and the mean values of Ct has been described.

Comment 13: Line 161 - The authors should explain how many samples were tested. It is stated that 35 milk and 25 swabs were tested the second time, whereas, according to data from line 97, there were only 53 samples (milk, swabs and carcass) collected the second time (?).

Line 165: As it was addressed in the previous comments: In point 2.1 “Sampling” the Authors present number of fetuses tested from the farm (4 animals), but no data about number or type of tested samples collected from this fetuses are given. The result “All 9 samples tested were positive” is non-informative, because it is unknown what type of samples were collected and tested and if they were collected from each fetus.

Line 167 – The Authors should clarify what 21 samples were tested - the data in lines 103-104 indicates that 1 fetus and 4 swabs were obtained for testing from this farm. It can be assumed that few samples were tested from fetus, but a difference of 15 samples seems too big.

Line 169 - As it was addressed in the previous comments: Number of tested samples (n=46) is incongruent with number presented in subsection “Samples”. In lines 105-106 it is stated that 2 fetuses and 33 milk samples were tested. Assuming that some internal organs of fetuses were tested as separate samples, it should be clarified and harmonised by the Authors.

Response 13: All the data have been corrected and the section have been rewritten to harmonise the informations.

Comment 14: Line 175 Table 1 – There is no reference to the table in the text. Were animals from farm A treated using drugs? If they were vaccinated, as it was mentioned in the text, the Authors should rephrase the heading of column to “Vaccination”. Moreover, it is not clear how many cows and sheep from Farm D were tested and were positive.

Response 14: The column name have been corrected

Comment 15: Line 201: The Authors should check if the number of reference is correct. It seems like it should be 20, not 18.

Line 207: The grammar should be corrected (e.g. has been done)

Response 15: They have been corrected both.

Comment 16: Line 256-258: The sentence is unclear and should be rephrased.

Response 16: The sentence has been rewritten.

Comment 17:Line 284: It is concluded that “ From 2019 to 2023 an increasing prevalence of C. burnetii was observed in this study”. Unfortunately, no results confirming this trend were present by the Authors in Result section. Only the calculations for 2023 were specified.

Response 17: The others data have been added, the sentences have been rewritten to better explane the prevalence.

Comment 18: 

Lines 327-330 – The Authors should consider combining this paragraph with sentence in lines 304-305 as it seems to be a continuation of the same thought.

Lines 331-332 – The Authors should consider moving this sentence to paragraph where the vaccination was discussed (lines 306-312).

References should be checked adjusted to the requirements of the Journal (e.g reference number 5 is given as: “ GACHE, K. and ROUSSET, E. and PERRIN, J. B. and DE CREMOUX, R. and HOSTEING, S. and JOURDAIN, E. and GUATTEO, 363...etc

Response 18: Done.

Reviewer 2 Report

Comments and Suggestions for Authors

The current manuscript is a retrospective review of C. burnetii (Cb) detection within the Abruzzo and Molise regional of Italy between 2019 and 2023. While the authors did report detection of Cb among wildlife species, the focus of the report was on 4 farms and the domestic animals on those farms, primarily goat, sheep and cattle. The most significant findings were the comparison of molecular detection methodologies and the ability to compare Cb prevalence pre and post vaccination on one of the farms. The authors found that molecular techniques are more reliable and sensitive than culturing this pathogen, which supports previous findings across the field, as well as demonstrating that Cb prevalence is often localized and broad country and sub-continent findings may not apply to locations within those regions.

The manuscript could use some moderate editing for clarity and language throughout. There are numerous run-on sentences in the introduction, and there are short paragraphs within the conclusions that are repetitive of concepts previously put forward. This distracts from the manuscript.

In summary, this is an interesting report that supports the need to develop formalized and consistent molecular detection techniques be employed when assessing Cb prevalence.  It emphasize the point that Cb rates can be quite localized and broad regional/country wide studies may not be indicative of local conditions relative to Cb. It will be of interest to the field when considering epidemiology of an outbreak as well as the considerations of background Cb prevelance.

Comments on the Quality of English Language

The current manuscript is a retrospective review of C. burnetii (Cb) detection within the Abruzzo and Molise regional of Italy between 2019 and 2023. While the authors did report detection of Cb among wildlife species, the focus of the report was on 4 farms and the domestic animals on those farms, primarily goat, sheep and cattle. The most significant findings were the comparison of molecular detection methodologies and the ability to compare Cb prevalence pre and post vaccination on one of the farms. The authors found that molecular techniques are more reliable and sensitive than culturing this pathogen, which supports previous findings across the field, as well as demonstrating that Cb prevalence is often localized and broad country and sub-continent findings may not apply to locations within those regions.

The manuscript could use some moderate editing for clarity and language throughout. There are numerous run-on sentences in the introduction, and there are short paragraphs within the conclusions that are repetitive of concepts previously put forward. This distracts from the manuscript.

In summary, this is an interesting report that supports the need to develop formalized and consistent molecular detection techniques be employed when assessing Cb prevalence.  It emphasize the point that Cb rates can be quite localized and broad regional/country wide studies may not be indicative of local conditions relative to Cb. It will be of interest to the field when considering epidemiology of an outbreak as well as the considerations of background Cb prevelance.

Author Response

Dear Reviewer,

thank you very much for your work, we have taken on board and integrated your suggestions. 

Commnet 1: The manuscript could use some moderate editing for clarity and language throughout. There are numerous run-on sentences in the introduction, and there are short paragraphs within the conclusions that are repetitive of concepts previously put forward. This distracts from the manuscript.

Response 1: The manuscript has been revised by removing sources of distraction and trying to better focus the topic.

Best regards,

Alessandra Alessiani

Reviewer 3 Report

Comments and Suggestions for Authors

The article, written by Alessiani et al., describes the epidemiological situation of Coxiella burnetii in some regions of central Italy. The article lacks scientific rigor in some sections. Other sections require important clarifications from the authors. Generally speaking, the results obtained would be worthy of being published but still need to be carefully reviewed. Below are some ideas for authors.

1) Title: The title does not correspond to what was done. There are no references to the molecular methods used, nor are there any references to the fact that the analyses were conducted on four farms. The number of farms is not sufficient to define the spread of Coxiella in different regions of Italy.

2) Abstract line 22: This is not necessarily correct because wild ruminants and other domestic ruminants (buffalo, llama, etc.) could be Q fever reservoirs. 

3) The abstract is missing the conclusions and implications of the present study.

4) Introduction: Some parts (such as lines 33–34) are not relevant to the purpose of the study.

5) The description of the herds in the form of a list could be supplementary material (including further information).

6) Further information on the molecular method used (e.g., target sequence, etc.) is necessary.

7) The results section, also divided by company, is rather confusing and does not highlight the results actually obtained.

8) Table: As point 7

9) The discussion is missing in piece of information regaring the presence of Q fever in ruminants in Italy. I suggest to the authors to include in the references the most relevant and recent study concerning seroprevalence and molecular prevalence studies performed in Italy (Sardinia, Piedmont, Campania, etc.).

10) The conclusion section is too long. Please move some sentences in the discussion

Comments on the Quality of English Language

English contains some minor errors that the authors could solve by themselves. 

Author Response

Dear Reviewer,

thank you for your valuable advice, the moscritus has been revised by accepting the suggestions of all the reviewers.

Comment 1: Title: The title does not correspond to what was done. There are no references to the molecular methods used, nor are there any references to the fact that the analyses were conducted on four farms. The number of farms is not sufficient to define the spread of Coxiella in different regions of Italy.

Response 1: The title has been modified to better fit with the article.

Comment 2:  Abstract line 22: This is not necessarily correct because wild ruminants and other domestic ruminants (buffalo, llama, etc.) could be Q fever reservoirs. 

Response 2: The sentence has been rewritten so that it is understood that these are animals of local interest and not the reservoir in general.

Comment 3/4: 

3) The abstract is missing the conclusions and implications of the present study.

4) Introduction: Some parts (such as lines 33–34) are not relevant to the purpose of the study.

Response 3/4: The text has been rewritten.

Comment 5: 5) The description of the herds in the form of a list could be supplementary material (including further information).

Response 5: The composition of the herds is shown in Table 1 and the adjacent text. We have no further information to add on this subject.

Comment 6: 6) Further information on the molecular method used (e.g., target sequence, etc.) is necessary.

Response 6: Informations have been added.

Comments 7/8:

7) The results section, also divided by company, is rather confusing and does not highlight the results actually obtained.

8) Table: As point 7

Response 7/8: The results section has been rewritten to better explane the data, harmonising the results expression.

Comment 9: 9) The discussion is missing in piece of information regaring the presence of Q fever in ruminants in Italy. I suggest to the authors to include in the references the most relevant and recent study concerning seroprevalence and molecular prevalence studies performed in Italy (Sardinia, Piedmont, Campania, etc.).

Response 9: Some data have been added. But it was difficult to find data comparable about molecular method or period of analysis.

Comment 10:10) The conclusion section is too long. Please move some sentences in the discussion

Response 10: the results section has been rewritten by eliminating superfluous parts and regrouping some elements to make the reading more harmonious.

Best regards,

Alessandra Alessiani

Round 2

Reviewer 1 Report

Comments and Suggestions for Authors

Thank you for carefully revising your paper addressing the reviewers’ comments. I think the manuscript has been improved by your efforts and it is acceptable for publication. I have one comment: Please check the spelling and correct "cattles", that is present in some lines of the manuscript, to proper plural form i.e. cattle.

Author Response

Dear Reviewer, thank you for your attention to our work, I have made the requested corrections.

Comment 1:  Please check the spelling and correct "cattles", that is present in some lines of the manuscript, to proper plural form i.e. cattle.

Response 1: Done, but some vocabulary says that cattle must be declined as cattle with plural verbs.